# Holmium-166 Transarterial Radioembolization for the Treatment of Intrahepatic Cholangiocarcinoma: A Case Series

**DOI:** 10.3390/cancers15194791

**Published:** 2023-09-29

**Authors:** Sim Vermeulen, Katrien De Keukeleire, Nicole Dorny, Isabelle Colle, Bert Van Den Bossche, Victor Nuttens, Dirk Ooms, Pieter De Bondt, Olivier De Winter

**Affiliations:** 1Nuclear Medicine Department, A.S.Z. Aalst, 9300 Aalst, Belgium; nicole.dorny@asz.be (N.D.); victor.nuttens@olvz-aalst.be (V.N.); dirk.ooms@olvz-aalst.be (D.O.); pieter.de.bondt@olvz-aalst.be (P.D.B.); olivier.de.winter@olvz-aalst.be (O.D.W.); 2Nuclear Medicine Department, OLV Aalst, 9300 Aalst, Belgium; 3Interventional Radiology Department, A.S.Z. Aalst, 9300 Aalst, Belgium; katrien.dekeukeleire@asz.be; 4Gastroenterology Department, A.S.Z. Aalst, 9300 Aalst, Belgium; isabelle.colle@asz.be; 5Abdominal Surgery Department, A.S.Z. Aalst, 9300 Aalst, Belgium; bert.vandenbossche@asz.be

**Keywords:** intrahepatic cholangiocarcinoma, transarterial radioembolization (TARE), selective internal radiation therapy (SIRT), holmium, safety

## Abstract

**Simple Summary:**

Intrahepatic cholangiocarcinoma (ICC) is a liver cancer with a poor prognosis, which is often not resectable at diagnosis. The search for novel effective treatments for ICC continues, but without much progress. One of the options is treatment with small particles carrying radionuclides yttrium-90 or holmium-166, which are delivered to the tumor through the artery to irradiate it locally. This allows for fewer side effects than systemic treatments while achieving good tumor control. While substantial evidence has accumulated for yttrium-90 radiotherapy, much less evidence is available for the more recent holmium-166. This work describes a series of seven patients with ICC treated with holmium-166 transarterial radiotherapy. Most of the patients achieved good tumor control after treatment and had no severe adverse events, highlighting the tolerability of the therapy. In cases of palliative intent of the treatment, this approach can help improve the quality of life and prolong the remaining years of life.

**Abstract:**

Background: Transarterial radioembolization (TARE) is used to treat primary and secondary malignancies in the liver that are not amenable to curative resection. Accumulating evidence demonstrates the efficacy and safety of TARE with yttrium-90 (^90^Y), which is the most widely used radionuclide for TARE, and later with holmium-166 (^166^Ho) for various indications. However, the safety and efficacy of ^166^Ho TARE in patients with intrahepatic cholangiocarcinoma (ICC) remains to be studied. Methods: This was a retrospective case series study of seven consecutive patients with ICC who were treated with ^166-^Ho-TARE in our center. We recorded the clinical parameters and outcomes of the TARE procedures, the tumor response according to mRECIST, subsequent treatments, and adverse events. Results: Three out of the seven patients had a partial or complete response. Two patients had stable disease after the first TARE procedure, and two of the patients (one with a complete response, and one with stable disease) were alive at the time of analysis. No serious adverse events related to the procedure were recorded. Conclusions: This is the first case series reporting the safety and tumor response outcomes of ^166^Ho-TARE for ICC. The treatment demonstrated its versatility, allowing for reaching a high tumor dose, which is important for improving tumor response and treating patients in a palliative setting, where safety and the preservation of quality of life are paramount.

## 1. Introduction

Intrahepatic cholangiocarcinoma (ICC) represents approximately 15% of primary hepatic malignancies and is characterized by high mortality, which has not decreased significantly over the years, despite the evolution of cancer treatment approaches [1]. It arises from the biliary epithelium located above the hilar junction [2]. Surgical resection is considered the only curative option; however, only 27–30% of patients are resectable at diagnosis, and even after a complete resection, most patients develop recurrent disease [3]. Systemic therapies, including targeted and immuno-therapies, are actively evolving, offering more treatment possibilities and improving outcomes for patients not eligible for curative treatments [1]. Recently, the TOPAZ study (NCT03785235) demonstrated significantly improved survival achieved by the addition of durvalumab to a gemcitabine and cisplatin (GemCis) standard treatment scheme [4]. The estimated 24-month survival rate offered by durvalumab reached 24.9% (compared to 10.4% using a placebo with GemCis), and the objective response rate was 26.7%. These positive results, however, remain limited.

Locoregional treatments for ICC can be used for liver-limited unresectable disease and include ablation, trans-arterial chemoembolization (TACE), transarterial radioembolization (TARE), hepatic arterial infusion (HAI) of chemotherapy, and external beam radiotherapy (EBRT) [5,6]. TACE takes advantage of the arterial blood supply to the tumor (as opposed to healthy liver parenchyma, which is alimented mainly through the portal vein) to locally deliver chemotherapy through the tumor-feeding artery either in the form of lipiodol emulsion or with drug-eluting microspheres. In both cases, the treatment is accompanied by artery embolization. Similarly to TACE, TARE is delivered intra-arterially, but the microspheres carry the therapeutic radionuclides and the anti-tumor action of TARE rely on the tumor’s irradiation, without a pronounced embolic effect [7]. This technique shows promising results for the treatment of ICC, even though the available studies are mostly small and include heterogeneous patient populations [8].

Currently, there are three types of therapeutic radioactive microspheres available for TARE. Two of them contain the radionuclide yttrium-90 (^90^Y), with a half-life of 64.1 h, and they are made of glass and resin, respectively. The most recent available product is based on holmium-166-containing poly-(L)-lactic acid (PLLA) microspheres. The radionuclide holmium-166 (^166^Ho) has a half-life of 26.8 h and, in contrast to ^90^Y, emits gamma radiation of 80.6 keV, allowing for imaging and quantification using SPECT/CT. Furthermore, the presence of gamma radiation permits the use of a low-activity (250 MBq) product, ^166^Ho-containing scout microspheres. These ^166^Ho scouts are identical to the therapeutic high-activity product and allow for a pre-treatment work-up and dosimetry simulation. A pre-treatment work-up is a necessary step for quantitative TARE planning, during which a surrogate substance with low activity, and which is suitable for imaging, is injected intra-arterially, simulating the TARE procedure. This allows for a personalized approach by calculating the expected radiation dose to be delivered to the tumor and the off-target radiation distribution to the healthy liver tissue and lungs. Scout microspheres have demonstrated superiority for predicting tumor doses in metastatic colorectal cancer (mCRC) patients and assessing potential lung shunt compared to technetium-99m (^99m^Tc)-macroaggregated albumin (MAA), routinely used for TARE treatment planning [9]. The importance of tumor dosimetry has been demonstrated with resin and glass ^90^Y microspheres, as it was the only factor besides the clinical variables that predicted increased outcomes in terms of tumor response and survival [10,11]. A recent multicenter retrospective study from Germany reported encouraging outcomes of the treatment of ICC with ^90^Y-TARE, especially in second-line and salvage settings [12].

For the more recent ^166^Ho-TARE, fewer clinical data are currently available. To date, several studies demonstrating the feasibility and good tolerability of ^166^Ho-TARE in primary and secondary liver malignancies have been published [13,14]. The dose–effect relationship has been demonstrated for ^166^Ho, with higher tumor-absorbed doses resulting in a higher tumor response rate and better survival, suggesting that tumor doses of at least 90 Gy should be used for mCRC patients [15,16]. One encouraging case report of ^166^Ho-TARE used for ICC has been published [17]; however, the evidence for ^166^Ho-TARE effects in this indication is currently limited. The purpose of this study is to report the clinical outcomes and safety of treatment of unresectable ICC with ^166^Ho-TARE in real-world clinical practice based on the retrospective data collection in our center. 

## 2. Materials and Methods

### 2.1. Study Population and Design

This was a retrospective case series data collection from ICC patients treated with ^166^Ho-TARE. All patients who were assigned to receive ^166^Ho-TARE by the institutional multidisciplinary tumor board and who received the treatment from April 2020 to November 2022 were included in this study. The inclusion criteria were >18 years old, a histological diagnosis of ICC; unresectable intrahepatic cholangiocarcinoma; performance status (PS) of 0–2; and a tumor size evaluable with mRECIST. The study included a total of 7 patients. The following data were collected: gender, age, metastases, performance status, tumor characteristics, date of diagnosis, aim of TARE (palliative or curative), TNM stage, number and types of previous treatments, tumor distribution (bilobar, lobar, or segmental), presence of extrahepatic disease, portal vein invasion, microvascular invasion, hepatic function parameters, presence of ascites or hepatic encephalopathy, treatment planning date, predicted target dose (in Gy), predicted non-target dose (in Gy), date of treatment, target volume (in mL), whole liver volume, administered activity (in GBq), tumor-absorbed dose, normal liver-absorbed dose and CA19.9 levels, tumor response at 3 and 6 months after the TARE, date of last follow-up or death, and date of progression. This retrospective registry was approved by the institutional ethics committee.

### 2.2. Study Procedures

Before radioembolization, the arterial perfusion of the lesions was monitored with diagnostic angiography, which included selective celiac and superior mesenteric arteriograms. A cone-beam CT (Philips Azurion 7 Biplane, Amsterdam, The Netherlands) was performed to determine the perfusion territory and prevent off-target deposition. In order to assess the vascularization of the tumor and the surrounding liver tissue and calculate the therapeutic radiation dose, the treatment planning procedure was performed before each treatment. This procedure was performed with ^166^Ho-containing scout microspheres, QuiremScout™ (Terumo Europe NV, Leuven, Belgium), by injecting 130 MBq for the right lobe, 70 MBq for the left lobe or lobar treatments, and 50 MBq for selective treatments. The scout microsphere distribution was visualized by SPECT/CT (Siemens Healthineers, Erlangen, Germany) imaging. Calculation of the treatment radiation dose was performed with dedicated software, QSuite^TM^ version 2.1 (Terumo Europe NV, Leuven, Belgium). A personalized dosimetry approach was taken for each patient, with the aim of achieving a tumor dose above 90 Gy in the largest lesion and a healthy liver dose below 40 Gy [18]. This was lowered, in the case of comorbidities (e.g., cirrhosis, liver fibrosis, previous partial hepatectomy), to 30–40 Gy. It was aimed to perform the radioembolization procedure with QuiremSpheres™ (Terumo Europe NV, Leuven, Belgium) within 7–14 days after the work-up. In most cases, the left and right lobes were treated separately in ICC patients with bilobar disease. The majority of the patients were kept overnight to monitor adverse events. Patients in good condition and injected selectively and below 5 GBq could be discharged after 6–8 h if the dose rate at 1 m was below 20 µSv/h. Patients returned 3–5 days after the treatment for SPECT/CT (Siemens Healthineers) imaging.

During the follow-up clinical visits, laboratory data collection and imaging were performed every 3 months after ^166^Ho-TARE until disease progression or death.

### 2.3. Outcome Measures

Adverse events (AE) were scored according to the Common Terminology Criteria for Adverse Events (CTCAE) version 5.0 [19]. Serious adverse events were defined as an AE grade ≥ 3, or any of the following AEs known to be associated with TARE, regardless of their CTCAE grading: acute pancreatitis, gastric ulceration, gastritis, radiation pneumonitis, radioembolization-induced liver disease (REILD), and cholecystitis. The tumor response was assessed using the modified response evaluation criteria (mRECIST) at 1, 3, and 6 months after ^166^Ho-TARE and was compared to pre-procedure imaging. A complete response (CR) was defined as the disappearance of any intratumoral arterial enhancement in all target lesions; a partial response (PR) was defined as a reduction in the sum of viable index lesion diameters by ≥30%; progressive disease (PD) was defined as a ≥20% increase in the viable lesions diameter or the appearance of new lesions; and stable disease (SD) was defined as any cases that did not qualify as a PR or PD.

## 3. Results

### 3.1. Case Series

A total of seven patients were enrolled in the study. All patients were male, with a median age of 73.7 years (range of 66–83) (Table 1).

The ECOG performance status was 0 in all but one patient. The intent of ^166^Ho-TARE was palliative in five patients and curative in two patients. Five patients had previously received systemic therapy (1–3 lines), and one of them also had external radiotherapy. One of the patients had portal vein invasion. Three of the patients had bilobar disease, two had segmental disease, and two had lobar disease. The TNM staging of the patients is presented in Table 1. 

A total of nine ^166^Ho-TARE procedures were conducted in the enrolled seven patients. The median delivered radiation activity per TARE was 4.36 GBq (range of 1.45–7 GBq), and the median absorbed dose per TARE in the target tumor was 106.6 Gy (range of 67–280 Gy). The median absorbed healthy liver dose was 32.9 Gy (range of 22–44 Gy), and in all but one patient, it did not exceed 40 Gy. The treatment approach was bilobar (*n* = 3), lobar (*n* = 4), or selective (*n* = 2).

One patient (patient N 4), who received a selective treatment with a radiation segmentectomy approach, achieved a CR, which was persistent during the following follow-up evaluations, and he was still alive at the time of the manuscript submission. Two patients had a PR, two patients had SD, and three patients had PD. The patients who had a PR and SD progressed at 3 or 6 months (Figure 1). 

Only one serious adverse event, namely radiation-induced cholecystitis, was registered in this case series during the follow-up after the ^166^Ho-TARE procedures. The most frequently reported adverse events after ^166^Ho-TARE were abdominal pain and fatigue.

### 3.2. Selected Case 1 (Patient N 4, Figure 2)

A 68-year-old male patient presented with a centrally located, enhancing hypervascular ICC of approximately 20 mm in diameter in segment 8 of the liver (Figure 2a). The patient had previously received systemic GemCis therapy (4 cycles). Cone-beam computed tomography confirmed the complete perfusion of the hypervascular tumor by the segment artery (Figure 2b). The ^166^Ho-TARE procedure was planned, with 5.24 GBq of activity that could be administered selectively, reaching a dose to the perfused area of 174 Gy and a tumor-absorbed dose of 280 Gy, with a healthy whole-liver dose of 22 Gy. The patient experienced no adverse events following the procedure. Follow-up MR imaging after 2 months and 4 months showed a complete response (according to mRECIST) of the tumor, with a decrease in the size of both the tumor and the resolution of tumoral hypervascularity. The patient started systemic treatment with FOLFOX because of increasing CA 19.9 levels but with stable findings with MRI. The patient was still alive 2 years later, with no evidence of recurrence of ICC. The presented case may serve as an example of successful ^166^Ho-TARE treatment when a high-tumor-absorbed dose could be achieved due to favorable blood alimentation of the tumor, which made selective treatment possible. Such a selective high-dose treatment appeared to be well-tolerated by the patient.

**Figure 2 cancers-15-04791-f002:**
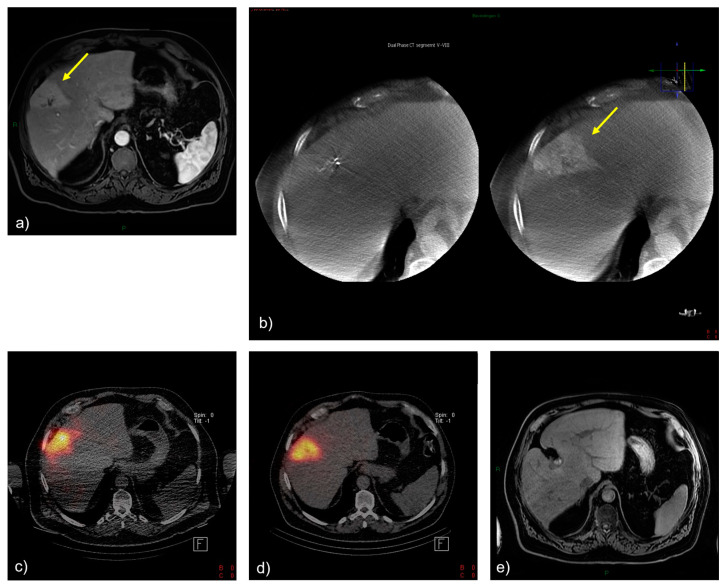
Treatment and tumor response after ^166^Ho-TARE of an ICCA in liver segment 8. (**a**) A pretreatment, contrast-enhanced, T1-weighted magnetic resonance (MR) image demonstrated a large, enhancing tumor in segment 8 (arrow). (**b**) Cone-beam computed tomography confirmed the complete perfusion of the hypervascular tumor by segment artery 8 (arrow). (**c**) Treatment planning with ^166^Ho scout. (**d**) Post-^166^Ho-TARE distribution verification. (**e**) Follow-up MR imaging at the last follow-up after transarterial radioembolization, showing a complete and persistent tumor response.

### 3.3. Selected Case 2 (Patient N 5, Figure 3)

An 83-year-old male patient presented with a centrally located ICC with smaller satellite lesions in segments 1, 3, and 7. MR imaging demonstrated heterogeneous enhancement, with central hypodensity. Procedural angiography confirmed the heterogeneous enhancement of the mass, with dual arterial supply from a left hepatic artery and a replaced right hepatic artery. The ^166^Ho-TARE procedure was planned, following a bilobar approach due to the presence of satellite lesions. The administered activity was 4.5 GBq, with a tumor-absorbed dose of 75 Gy, and a normal liver dose of 38 Gy. The patient experienced no adverse events following the procedure. Follow-up MR imaging 3 months after the procedure demonstrated a partial response according to mRECIST, and the majority of the satellite lesions appeared to be stable. Due to the evolution of the satellite lesions in segment 4 and the patient’s preference to have TARE over systemic chemotherapy, the patient received a second selective ^166^Ho-TARE treatment 4.5 months later. The patient experienced radiation-induced cholecystitis after the second ^166^Ho-TARE procedure due to microsphere deposition in the gallbladder wall. At the last consultation 3.5 months later, the patient found his quality of life optimal, did not receive systemic chemotherapy, and did not find further medical follow-up necessary. The patient died 3 months later, presumably due to clinical deterioration. In this case, the presence of satellite lesions precluded selective ^166^Ho-TARE with a high tumor dose, and only partial success was achieved in terms of the tumor response. However, ^166^Ho-TARE may have contributed to maintaining the patient’s quality of life and decreased his need for chemotherapy in the palliative setting.

**Figure 3 cancers-15-04791-f003:**
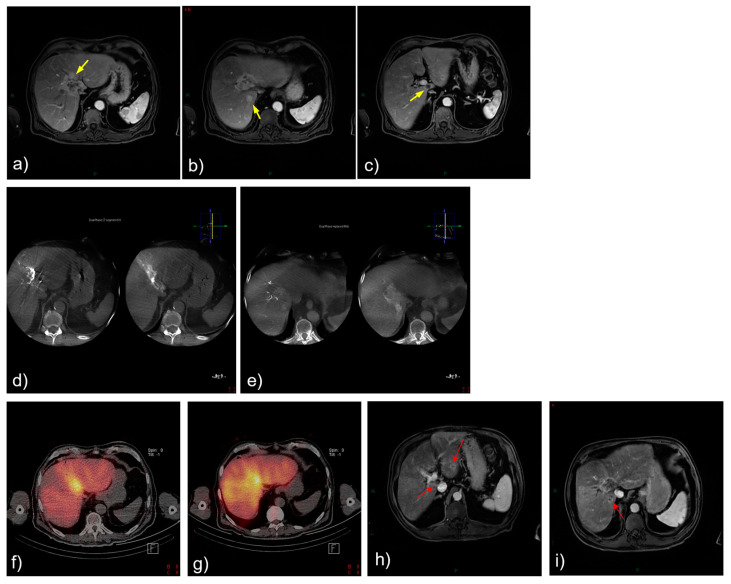
Treatment and tumor response after ^166^Ho-TARE of an ICCA in the dome involving liver segments 8 and 4, with additional small satellite lesions in 1, 3, and 7. (**a)**–(**c**) Pretreatment, contrast-enhanced, T1-weighted magnetic resonance (MR) image demonstrated heterogeneous enhancement, with central hypodensity (arrows). (**d**,**e**) Procedural angiography confirmed the heterogeneous enhancement of the mass with dual arterial supply from a left hepatic artery and a replaced right hepatic artery. (**f**) Treatment planning and (**g**) treatment with ^166^Ho-TARE with lobar approach. (**h**,**i**) Follow-up MR imaging 3 months after transarterial radioembolization demonstrated partial response, despite baseline hypovascularity of the mass but also the satellite lesions (red arrows).

## 4. Discussion

In this study, we present a series of cases of ICC treated with ^166^Ho-TARE at our center. Our experience confirms the feasibility of the ^166^Ho-TARE procedure with ^166^Ho scout treatment planning in these patients. The observed outcomes are in line with the results previously reported for ^90^Y-TARE treatments, with a median OS between 6 and 22 months [20]. In a recent study reporting real-world evidence outcomes of ICC treatment with ^90^Y-TARE, an mOS of 12 months was observed for patients receiving the treatment as a first line, while mOS of 11.8 and 8.4 months were observed for patients receiving radioembolization as second-line and salvage treatments, respectively [12]. The small sample size and heterogeneity of the patients included in this case series did not allow for any statistical comparisons of our data with previously published results; however, our observations lie within the published ranges.

^166^Ho-TARE appeared to be well tolerated, as no serious adverse events were recorded. The absorbed healthy liver radiation dose did not exceed 40 Gy in all but one patient, which may explain the absence of recorded REILD cases in this series. Previously, a positive association between the parenchymal dose of ^166^Ho and toxicity was described in mCRC patients [16]. Because of the predominantly palliative use and the absence of a dose–response relationship in ICCA for ^166^Ho, a lower parenchymal dose (median of 30.77 Gy; range of 18–44 Gy) was chosen over the maximum proposed parenchymal dose of 55 Gy in mCRC. The main side effects were abdominal pain and fatigue, as also reported for post-embolization syndrome associated with the use of ^90^Y-TARE [21].

This study also showed the possibilities of achieving local tumor control with ^166^Ho-TARE, since 3 out of 6 patients had an objective response at 1 month after treatment, although the majority of the patients had disease progression later because of their palliative stage and the aggressive nature of their disease. Overall, the clinical outcomes observed in this case series were in agreement with the previously reported results for ^166^Ho-TARE [13,14,17,22]. It has previously been demonstrated that a selective approach to TARE allows for improved treatment outcomes. However, a selective approach is not always feasible due to the tumor load and/or distribution and the specifics of the vascular access. In future studies, it would be interesting to focus on the efficacy of ^166^Ho-TARE for ICC when stringent inclusion criteria are applied to select the patients who may benefit the most from the treatment.

Locoregional treatments, including TARE, appear to be a valuable option for the palliative treatment of patients with unresectable ICC, which may allow for avoiding or delaying the use of systemic therapies associated with substantial toxicity, and therefore, potentially improving the patient’s quality of life [12,23]. Moreover, accumulating evidence highlights the possibilities of combining locoregional and systemic treatments. A recently published DELTIC trial (NCT01648023) reported a significant improvement in clinical outcomes in patients treated with TACE, a combination of systemic GemCis and locoregional (irinotecan drug-eluting beads, DEBIRI) treatments [24]. In this study, a median progression-free survival (PFS) of 31.9 (95% CI 8.5–75.3) and a median OS of 33.7 (95% CI 13.5–54.5) months were reported in the combination group. 

The advantage of ^166^Ho microspheres used for TARE is the possibility of both SPECT and MRI imaging, allowing for a more accurate assessment of the microsphere distribution during treatment planning [25]. This case series adds to the knowledge of ^166^Ho-TARE use for ICC, as very limited evidence for this technique in ICC treatment has been published so far. Currently, a tumor dose of >150 Gy is recognized as a desirable threshold to achieve improved outcomes in primary liver tumors [18,26]. In our cohort, the high > 150 Gy tumor dose resulted in a sustained CR, but an ORR could also be observed at lower tumoral doses. Furthermore, another group demonstrated that in the neoadjuvant setting, a tumor dose of 88 Gy resulted in tumor downstaging to surgery [17]. Therefore, the balance among the efficacy, treatment objective, technical feasibility, and patient profile should be taken into consideration. In future studies, further refinement of the tumor and healthy liver dose–response relationship should be available.

This study was limited by its retrospective single-center design and the small number of patients included. However, these preliminary data suggest that ^166^Ho-TARE can be regarded as a viable option for the treatment of unresectable ICC, with a promising potential to achieve local stable disease and good tolerability.

## 5. Conclusions

This is the first case series study reporting the outcomes of the treatment of patients with unresectable ICC with ^166^Ho-TARE. The results of this case series suggest that ^166^Ho-TARE can be used for the treatment of such patients, allowing for the achievement of good disease control with good tolerability. The ^166^Ho-TARE technique offers the versatility of being used both in the setting of selective treatment with a high targeted tumor dose, aimed at achieving tumor response, and in the palliative salvage setting, aimed at preserving the patient’s life duration and quality. Further studies with a larger number of patients and a comparative design are needed to confirm the therapeutic potential of ^166^Ho-TARE in this patient population.

## Figures and Tables

**Figure 1 cancers-15-04791-f001:**
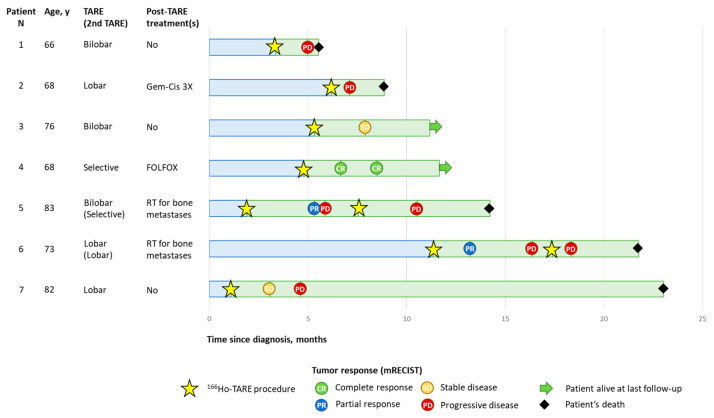
Patients’ disease course. Tumor response evaluated according to mRECIST. RT, radiotherapy; TARE, transarterial radioembolization.

**Table 1 cancers-15-04791-t001:** Patient and treatment characteristics.

No.	Sex	Age at Diagnosis (Years)	PS	Comorbidities	Tumor Distribution	T-N-M	Systemic Treatment before TARE	Target Volume (mL)	N of TARE Procedures	Administered Activity (GBq)	Tumor Dose (Gy)	Normal Liver Dose (Gy)
1	Male	66	0	AHT, DM2, asthma, metastatic esophageal cancer	Bilobar	2-0-0	Carbotaxol, Cis-5FU	1845	1	7	71	44
2	Male	68	0	DM2	Lobar	4-1-1	Cisplatinum–gemcitabine	2787	1	5.6	80	23
3	Male	76	0	AHT, DM2, iCVA, HF with reduced ejection fraction, pacemaker	Bilobar	4-1-0	Gemcitabine–oxaliplatin	1660	1	4.26	67	34
4	Male	68	0	AHT, DM2, iCMP, OSAS	Segmental	1-1-0	Cisplatinum–gemcitabine	451	1	5.24	280	22
5	Male	83	2	AHT, colon cancer, mechanical AVP	Bilobar	2-1-1	None	1693	2	4.5	75	38
6	Male	73	0	AHT, DM2, OSAS	Lobar	3-1-1	Cisplatinum–gemcitabine, FOLFOX	1680	2	4.94	70	31
7	Male	82	0	HF with reduced ejection fraction	Segmental	4-0-0	None	1397	1	4.22	103	38

AHT, arterial hypertension; AVP, aortic valve prosthesis; DM2, diabetes mellitus type 2; HF, heart failure; iCMP, ischemic cardiomyopathy; iCVA, ischemic cerebrovascular accident; OSAS, obstructive sleep apnea syndrome; PS, performance status; TARE, transarterial radioembolization; T-N-M, tumor, lymph node, metastasis (TNM) staging system.

## Data Availability

The data presented in this study are available upon request from the corresponding author. The data are not publicly available due to hospital policy.

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
