# Peer review of "Holmium-166 Transarterial Radioembolization for the Treatment of Intrahepatic Cholangiocarcinoma: A Case Series"

_cancers, 2023, doi:10.3390/cancers15194791_

Round 1

Reviewer 1 Report

The use of transarterial radioembolization with holmium-166 for the treatment of intrahepatic cholangiocarcinoma is discussed in this manuscript. The article seems to be well organized, including the study's background, methodology, findings, and conclusion. Nonetheless, if you want to improve the quality of the article, you could think about the following suggestions and experiments:

1:Improve the introduction's content to include intrahepatic cholangiocarcinoma (ICC), its available therapies, and the necessity for further therapies such transarterial radioembolization. This will help in establishing the context for the study's significance.

2:Regarding patient demographics, consider about including more information, such as age ranges, gender distributions, and any pertinent comorbidities. This may provide a more accurate image of the research patient sample. Provide the selection criteria for the particular dosages and treatment modalities. Explain the thinking behind the dosage choice, in particular how to manage the tumor while causing the least amount of harm to healthy tissue.

3: Examine if specific patient traits—like tumor size, location, or particular genetic markers—correlate with better treatment results. This can include looking through the any possible indicators of treatment response.

4:Enhancing treatment responses and lowering adverse occurrences may result from investigating the dosimetry analysis.

5: Compare the results of Holmium-166 TARE with those of other ICC therapies that are currently being used, such as radiation therapy, chemotherapy, or surgical resection. A retrospective investigation or even a thorough review and meta-analysis may fall under this category.

6: Investigating the underlying processes of Holmium-166's impact on tumor cells by working with a lab or doing in vitro tests. This can include researching modifications in gene expression, DNA damage repair, or post-treatment cell cycle progression.

No any

Author Response

The use of transarterial radioembolization with holmium-166 for the treatment of intrahepatic cholangiocarcinoma is discussed in this manuscript. The article seems to be well organized, including the study's background, methodology, findings, and conclusion. Nonetheless, if you want to improve the quality of the article, you could think about the following suggestions and experiments:

We thank the Reviewer for his/her time and careful evaluation of our manuscript. We took effort to address the specific point as fully as possible: please see the answers below. We hope that the amended version of the manuscript will be found suitable for publication.

1:Improve the introduction's content to include intrahepatic cholangiocarcinoma (ICC), its available therapies, and the necessity for further therapies such transarterial radioembolization. This will help in establishing the context for the study's significance.

We amended the Introduction by adding more relevant references on the TARE use for the treatment of ICC, including recently published studies. However, we deem sufficient the amount of information we provide for ICC as disease, since the focus of this manuscript is the use of specific treatment technique for this indication. We do refer the reader to the relevant papers providing exhaustive information on ICC.

2:Regarding patient demographics, consider about including more information, such as age ranges, gender distributions, and any pertinent comorbidities. This may provide a more accurate image of the research patient sample. Provide the selection criteria for the particular dosages and treatment modalities. Explain the thinking behind the dosage choice, in particular how to manage the tumor while causing the least amount of harm to healthy tissue.

Following the Reviewer’s suggestion, we added information on patients’ comorbidities in the baseline characteristics Table. In the Table, and in the Results section, we also report the available information on patients’ demographics and background characteristics. We also include information on the dosage received. We include notes on the dosage choice in the Discussion section.

3: Examine if specific patient traits—like tumor size, location, or particular genetic markers—correlate with better treatment results. This can include looking through the any possible indicators of treatment response.

We thank the Reviewer for this suggestion. However, the small patient number included in this study precluded any proper correlation analyses or evaluation of possible indicators of treatment response. However, we took note of this suggestion, and will perform such analyses in the future, once we have a sufficient patient population size in our center.

4: Enhancing treatment responses and lowering adverse occurrences may result from investigating the dosimetry analysis.

Similarly to the previous point: the small patient number did not allow us to evaluate dose-response or dose-toxicity, but we may use the collected data for future, larger retrospective studies evaluating dosimetry.

5: Compare the results of Holmium-166 TARE with those of other ICC therapies that are currently being used, such as radiation therapy, chemotherapy, or surgical resection. A retrospective investigation or even a thorough review and meta-analysis may fall under this category.

We thank the Reviewer for this suggestion. We included a reference to a very recent article evaluating TARE with Y90 for ICC treatment. We also refer the reader to the relevant studies with locoregional and systemic therapies for ICC in the Introduction and Discussion sections. However, the small sample and the design of the current study precludes any comparative evaluations.

6: Investigating the underlying processes of Holmium-166's impact on tumor cells by working with a lab or doing in vitro tests. This can include researching modifications in gene expression, DNA damage repair, or post-treatment cell cycle progression.

Thank you for this interesting suggestion! We believe, however, that such in vitro study falls beyond the scope of the current publication, which was aimed to report a selection of cases. However, we added a reference to the SIRT mechanism of action in the Introduction.

Reviewer 2 Report

This paper is dealing with Holmium-166 Transarterial Radioembolization for the treat- 2 ment of Intrahepatic Cholangiocarcinoma: A Case Series. The paper is well written and well organized and contains results that could be interesting for some readers. However, there are some shortcomings that should be resolved.

1)      Graphical abstract is not clearly visible.

2)      On Page 6, an empty line is needed between the caption of Figure 1 and the paragraph.

3)      On Page 7, an empty line is needed between the caption of Figure 2 and the paragraph.

4)      The caption of Figure 3 is not clear. It is too long. In  Figure, I cannot see labels a,b,c,d,e,  f, and g.

5)      The conclusion should be rewritten. It is too short.

6)      The list of references contains only 24 references. Some important references are missing and have to be added.

Author Response

This paper is dealing with Holmium-166 Transarterial Radioembolization for the treat- 2 ment of Intrahepatic Cholangiocarcinoma: A Case Series. The paper is well written and well organized and contains results that could be interesting for some readers. However, there are some shortcomings that should be resolved.

We thank the Reviewer for the positive evaluation of our manuscript. We addressed all the specific points listed below and hope for the positive final evaluation.

  • Graphical abstract is not clearly visible.

We will ensure the quality of the graphical abstract in the final submission

  • On Page 6, an empty line is needed between the caption of Figure 1 and the paragraph.

Corrected

  • On Page 7, an empty line is needed between the caption of Figure 2 and the paragraph.

Corrected

  • The caption of Figure 3 is not clear. It is too long. In  Figure, I cannot see labels a,b,c,d,e,  f, and g.

Corrected

  • The conclusion should be rewritten. It is too short.

We amended the Conclusion part following this and the other Reviewer’s comments

  • The list of references contains only 24 references. Some important references are missing and have to be added.

We added more relevant references to the list, also following the other Reviewer’s comments. We are open for proposals of other relevant references that we might have missed.

Round 2

Reviewer 1 Report

All of the reviewer's comments have been addressed by the authors.

No any